# Minimal change disease following COVID-19 vaccination: A systematic review

**Konstantinos S. Kechagias**[1]*, **Joshua D. Laleye**[1], **Jan Drmota**[1],
**Georgios Geropoulos**[2], **Georgios Kyrtsonis**[1], **Marina Zafeiri**[3], **Konstantinos Katsikas Triantafyllidis**[4], **Dimitra Stathi**[5]

1 Department of Metabolism, Digestion and Reproduction, Faculty of Medicine, Imperial College London, London, United Kingdom, 2 Western General Hospital, Lothian NHS, Edinburgh, United Kingdom, 3 School of Cardiovascular Medicine and Sciences, King's College London, London, United Kingdom, 4 Department of Nutrition and Dietetics, Royal Marsden NHS Foundation Trust, London, United Kingdom, 5 Department of Endocrinology and Diabetes, King's College Hospital NHS Foundation Trust, London, United Kingdom

* konstantinos.kechagias18@imperial.ac.uk

**Data Availability Statement:** All relevant data are within the manuscript and its Supporting Information files.

## Abstract

### Background

The newly developed COVID-19 vaccines are highly effective and safe. However, a small portion of vaccine recipients experience a wide range of adverse events. Recently, glomerular disease, including the development of Minimal Change Disease (MCD), has been observed after administration of different COVID-19 vaccines, although causality remains a matter of debate.

### Aim

The aim of this systematic review was to comprehensively examine the available literature and provide an overview of reported cases of MCD following vaccination against SARS-CoV-2.

### Results

We identified 46 eligible articles which included 94 cases with MCD following COVID-19 vaccination of which one case was reported twice due to a second relapse. Fifty-five participants were males (59.1%, 55/93) and 38 (40.9%, 38/93) were females with a mean age of 45.02 years (SD:20.95). From the included patients 50 (50/94, 53.1%) were described as new-onset and 44 (46.9%, 44/94) as relapse. On average, symptomatology developed 16.68 days (SD: 22.85) after the administration of the vaccine irrespective of the dose. Data about symptoms was reported in 68 cases with the most common being oedema (80.8%, 55/68), followed by weight gain (26.5%, 18/68) and hypertension (16.1%, 11/68). In terms of outcome, more than half of the patients went into remission (61%, 57/94), while 18 recovered or improved post treatment (19.1%, 18/94). Two people relapsed after treatment (2.1%, 2/94) and two cases (2.1%, 2/94) were reported as not recovered.

**Funding:** The authors received no specific funding for this work.

**Competing interests:** The authors have declared that no competing interests exist.

## Conclusion

MCD is possibly a condition clinicians may see in patients receiving COVID-19 vaccines. Although this adverse event is uncommon, considering the limited published data and the absence of confirmed causality, increased clinical awareness is crucial for the early recognition and optimal management of these patients.

## Introduction

In late 2019, a global pandemic, which created extraordinary socio-economic consequences, emerged due to an outbreak of an uncommon viral pneumonia [1–3]. The aetiological factor was later identified as a previously unknown strain of coronavirus named Severe Acute Respiratory Syndrome Coronavirus 2 (SARS-CoV-2), responsible for the onset of coronavirus disease 2019 (COVID-19). The disease has since spread extensively, impacting hundreds of millions of individuals across the globe [4, 5].

Various vaccines have been utilised successfully against SARS-CoV-2 such as COMIR-NATY (BioNTech-Pfizer's COVID-19 mRNA vaccine BNT162b2), COVID-19 Vaccine Moderna (Moderna's mRNA vaccine-1273), VAXZEVRIA (AstraZeneca-Oxford University's ChAdOx1-nCoV19), COVID-19 Vaccine Janssen (Janssen's Ad26.COV2.S) and CoronaVac COVID19 vaccine (Sinovac Biotech's Vero cell) [6, 7]. Currently, nearly two-thirds of the global population have received at least one dose of a COVID-19 vaccine, with more than 13 billion doses administered worldwide [8].

A plethora of published studies have demonstrated the safety and effectiveness of the aforementioned vaccines, with only infrequent adverse events reported in the literature [9–12]. Nonetheless, isolated adverse reactions after COVID-19 vaccine administration are unavoidable, given the vast amount of vaccination doses needed to curb the spread of COVID-19 [13, 14]. At present, patients commonly experience various reported adverse symptoms, such as muscle pain, fever, headache, nausea, and vomiting. In addition to the frequently observed adverse effects following COVID-19 vaccination, patients have also reported a wide range of complaints and symptoms, including immune-mediated adverse events [13, 15–18].

Recently, there is a growing number of reports regarding the development of Minimal Change Disease (MCD) in patients following their initial or second COVID-19 vaccine doses; However, these cases have not yet undergone thorough investigation, and the administration of COVID-19 vaccines has not been recognised as a causative factor for renal dysfunction. To address this gap, our study systematically analysed the existing literature to present a comprehensive summary of documented cases of MCD following SARS-CoV-2 vaccination.

## Methods

This review was reported based on the "Preferred Reporting Items for Systematic Reviews and Meta-Analyses" (PRISMA) guidelines (S1 Fig).

### Literature search

Two reviewers (KSK, JDL) searched PubMed and Scopus library databases from inception until January 2023 independently. The search included the following terms: "(COVID 19 vaccin* OR SARS-COV2 vaccin*) AND (minimal change disease OR glomerulonephritis OR nephrotic OR nephritic)". There were no limitations placed regarding study design,

geographic region, or language. Additionally, a manual search of references cited in the included articles and relevant published reviews was conducted to identify any missed studies. Discrepancies during the literature search were resolved by a third investigator (DS).

### Eligibility criteria

We included studies that provided data for new onset or relapse of MCD following COVID-19 vaccination with at least one dose. All study designs were considered eligible for inclusion. Review articles, abstracts submitted in conferences and non-peer reviewed sources were not eligible for inclusion. Studies on in vitro and animal models were excluded.

### Data extraction and handling

In all studies, patient data was retrieved and handled by two authors (JDL, JD) who conducted the data extraction independently. We collected the following information: sex, age, comorbidities, vaccine type, number of doses received, presenting complains and symptoms, history of previous COVID-19 infection, laboratory tests including antibodies, primary diagnosis, imaging findings, therapeutic management and clinical outcome. Any disagreements were discussed and resolved by a third author (KSK).

### Quality assessment

The quality of the included studies was assessed using the criteria established by the Task Force for Reporting Adverse Events of the International Society for Pharmacoepidemiology (ISPE) and the International Society of Pharmacovigilance (ISoP) [19]. The evaluation was based on the satisfactory reporting of 12 different elements, including the title, patient demographics, current health status, medical history, physical examination, patient disposition, drug identification, dosage, administration/drug reaction interface, concomitant therapies, adverse events, and discussion. Each element was assigned a score of either 0 (lack of information) or 1 (information present) for the studies.

## Results

### Study characteristics

The initial literature search yielded 830 publications. In the first screening 777 studies were excluded as irrelevant. Forty-six studies [20–65] were found to be eligible for the systematic review based on the predefined inclusion criteria (Fig 1). Twenty of the studies were conducted in Asia, 16 in Europe, 9 in Americas, and 1 in Australia. Seven studies were case series and 39 were case reports (Table 1).

We identified a total of 94 cases of MCD following COVID-19 vaccination, of which one case was reported twice after relapsing following the second dose.

Fifty-five participants were males (59.1%, 55/93) and 38 (40.9%, 38/93) were females with a mean age of 45.02 years (SD:20.95). From the included patients 50 (50/94, 53.1%) were characterised as new-onset and 44 (46.9%, 44/94) as relapse. The mean age of individuals with MCD relapse was 41.6 (SD:20). For most of the patients (79.5%, 74/93) data regarding COVID-19 infection before or at the time of diagnosis was not provided. Among the remaining patients only 2 were previously infected with SARS-CoV-2. In 2 cases, vaccine brand was not reported (2.1%, 2/94). The majority of the patients received COMIRNATY (58.5%, 55/94), followed by COVID-19 Vaccine Moderna (20.2%, 19/94) and VAXZEVRIA (14%, 13/94), while 4 participants received COVID-19 Vaccine Janssen (3.2%, 3/94) and CoronaVac (1%, 1/94). In one case vaccine type was reported as modRNA (1%, 1/94). The majority of patients developed

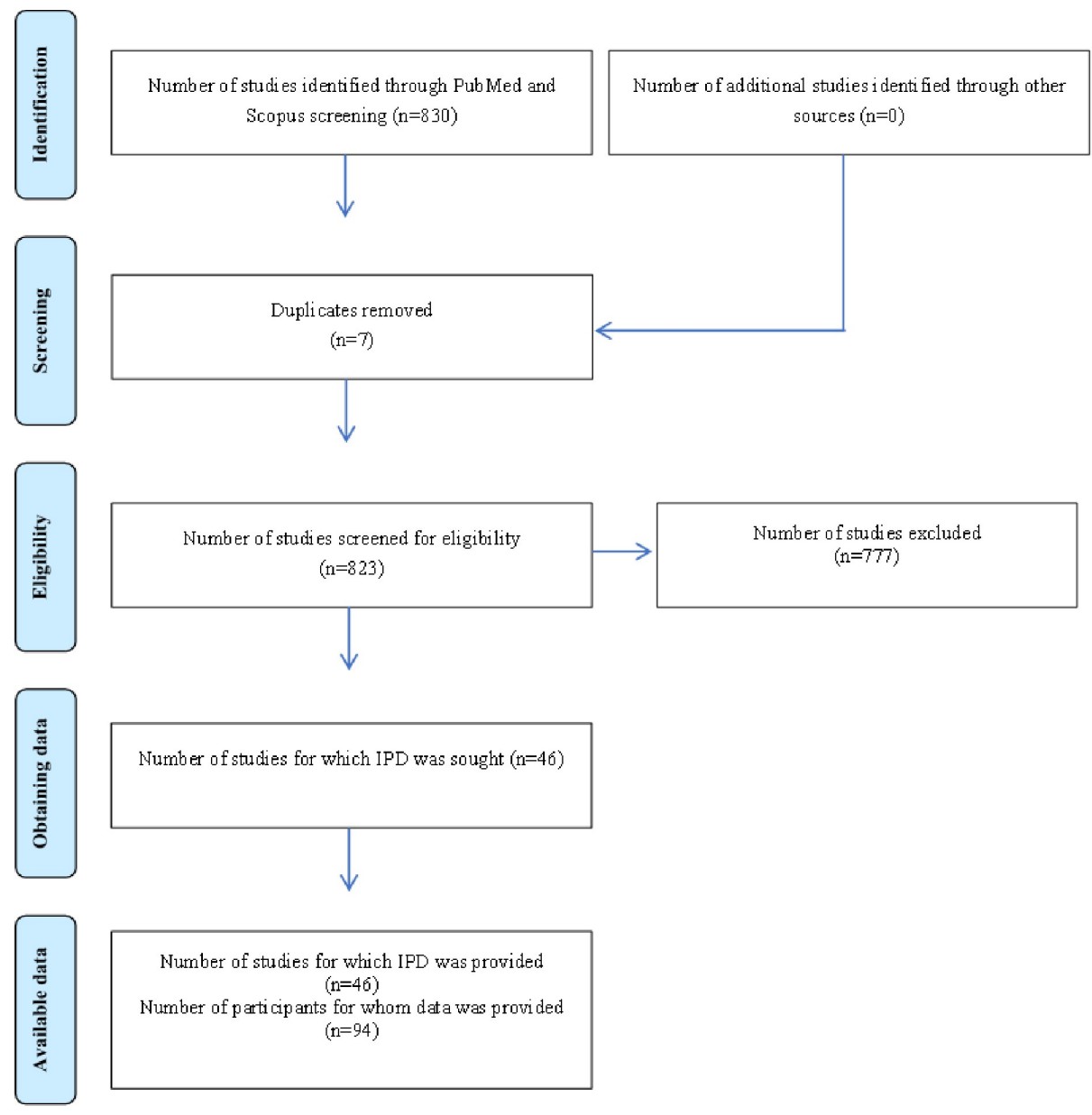

**Fig 1. PRISMA flowchart.** IPD: individual patient data.

symptoms after the first dose (55.5%, 52/94), followed by the second dose (39.3%, 37/94), third dose (2.2%, 2/94), booster (1%, 1/94), both first and second doses (1%, 1/94), while in one case relevant data was not provided (1%, 1/94).

On average, the symptoms developed 16.68 days (SD: 22.85) after the administration of the vaccine irrespective of the dose. Data about symptomatology was reported in 68 individuals with the most common symptom being oedema (80.8%, 55/68), weight gain (26.5%, 18/68) and hypertension (16.1%, 11/68). MCD was confirmed with biopsy in 76 cases (80.8%, 76/94). Sixteen cases (17%, 16/94) were relapses and biopsy was not repeated. In two cases (2.1%), diagnosis was based on clinical suspicion (S1 Table). The majority of patients received steroids (91.5%, 86/94), while some patients were treated with immunosuppressive agents (22.3%, 21/94) and diuretics (17%, 16/94). More than half went to remission (61%, 57/94), while 18

**Table 1. Characteristics of the included studies.**

| Author, Year, Country | Case number, Age, Gender | Comorbidities | Previous glomerulopathy | Previous COVID-19 infection | COVID-19 vaccine type and dose | New onset/ relapse of minimal change disease post vaccination | Main presenting symptoms | Days for the onset of symptoms | Treatment | Outcome |
|---|---|---|---|---|---|---|---|---|---|---|
| Marampudi 2022 USA | case 1 54 F | Hypertension | MCD | none | mRNA (Pfizer), first | relapse | • Lower Limb Oedema • Foamy Urine | 7 | Prednisolone (50mg/day) | prednisone taper rituximab if relapses |
| Pella 2022 Greece | case 1 18 M | None | None | none | mRNA (Pfizer), first | new | • Nausea • Bloating • Abdominal Pain • Lower Limb Oedema • Weight Gain | 11 | Irbesartan 150 mg Methylprednisolone 48 mg | Complete remission in 6 weeks |
| Alhosaini 2022 UAE | case 1 16 M | None | n/a | NA | mRNA (Pfizer), second | new | • Lower Limb Oedema • Ankle Swelling • Abdominal Pain • Weight gain | 7 | Prednisone 60mg, furosemide, Olmesartan | Oedema resolved after 1 week |
| Mochizuki 2022 Japan | case 1 25 F | None | None | NA | mRNA (Moderna), first | new | • Facial Oedema • Peripheral oedema • Weight gain | 26 | IV Methylprednisolone 500 mg/day for 3 days Oral Prednisolone 45mg/day. | Complete remission by day 10 |
| Park 2022 Korea | case 1 34 M | NA | None | NA | mRNA (Moderna), second | new | • Peri-ocular oedema • Dyspnoea • Weight gain | 3 | Prednisolone | Remission after 11 weeks |
| | case 2 60 M | NA | None | NA | mRNA (Moderna), second | new | • Oedema • Weight gain | 5 | Steroids | Complete remission after 2 weeks |
| Hartley 2022 UK | case 1 80s F | None | None | NA | mRNA (Pfizer), first | new | • Oedema • Reduced urine output • Hypertension | 2 | Loop diuretics, Low molecular weight heparin, Steroids, Levothyroxine | Complete remission |
| | case 2 40s M | Wolf-Parkinson-White Syndrome, Cardiac ablation | MCD | NA | mRNA (Pfizer), first | relapse | • Oedema • Diarrhoea • Vomiting | 1 | Furosemide, Prednisolone, Ciclosporin | Complete remission |
| Leong 2021 Singapore | case 1 42 F | None | MCD | NA | mRNA (Moderna), second | relapse | • Oedema • Frothy urine | 11 | Prednisolone | Remission within 2 weeks |
| | case 2 30 M | None | MCD | NA | mRNA (Pfizer), second | relapse | • Oedema • Frothy urine | 7 | Prednisolone | Remission within 2 weeks |
| Tanaka 2022 Japan | case 1 69 F | Hypertension, Hyperlipidaemia | None | NA | mRNA (Pfizer), second | new | • Oedema • Weight gain | 9–18 days | Prednisolone | Complete remission within 1 month |
| Jongvilaikasem 2022 Thailand | case 1 14 M | None | None | NA | mRNA (Pfizer), first | new | • Oedema • Hypertension | 5 | Corticosteroids | Partial remission after 5 weeks treatment |
| Marinaki 2021 Greece | case 1 55 F | Hypothyroidism | None | NA | mRNA (Pfizer), first and second | mode (after second dose) | • Oedema • Weight gain | 4 days after first dose. 1 day after second | Prednisolone | Remission after 10 days |
| Biradar 2021 India | case 1 22 M | None | None | NA | Viral Vector (Oxford-Astrazeneca), first | new | • Oedema | 11 | Prednisolone | Remission after 1 week |
| Unver 2021 Turkey | case 1 67 F | Type 2 diabetes mellitus | None | NA | Inactivated (Sinovac), first | new | • Oedema • Weight gain | 20 | Ramipril, Valsartan, Nebivolol, Rosuvastatin, Furosemide | Represented after second dose |
| Lebedev 2021 Israel | case 1 50 M | None | None | NA | mRNA (Pfizer), first | new | • Oedema • Abdominal distension | 4 days | Prednisolone | Improved 17 days later |
| Hanna 2021 Canada | case 1 60 M | None | None | NA | mRNA (Pfizer), first | new | • Oedema • Fatigue • Shortness of breath on exertion | 10 | Ramipril, Furosemide, Prednisolone | Remission from 14 days confirmed 3 weeks later |

*(Continued)*

**Table 1.** (Continued)

| Author, Year, Country | Case number, Age, Gender | Comorbidities | Previous glomerulopathy | Previous COVID-19 infection | COVID-19 vaccine type and dose | New onset/ relapse of minimal change disease post vaccination | Main presenting symptoms | Days for the onset of symptoms | Treatment | Outcome |
|---|---|---|---|---|---|---|---|---|---|---|
| Baskaran 2022 Australia | case 1 31 F | NA | None | NA | mRNA (Pfizer), second | new | • Oedema | 21 | High-dose steroids | Good response to treatment |
| | case 2 55 M | NA | None | NA | Viral Vector (Oxford-Astrazeneca), second | new | • Oedema • Ascites | 7 | Prednisolone | Improved kidney function and proteinuria |
| Thappy 2021 Qatar | case 1 43 M | None | None | none | mRNA (Moderna), first | new | • Oedema • Dyspnoea | 7 | Furosemide, Amlodipine, Prednisolone | No oedema, raised serum albumin, reduced urine protein after 2 weeks |
| Abdulgayoom 2021 Qatar | case 1 45 F | Hypothyroidism, Atopic dermatitis, Heterozygous factor V mutation | None | NA | mRNA (Pfizer), first | new | • Oedema • Abdominal distention • Foamy urine • Abdominal ascites | 4 | Furosemide, Prednisolone, Vitamin D, Calcium, Pantoprazole, Trimethoprim/ Sulfamethoxazole | NA |
| Klomjit 2021 USA | case 1 83 M | NA | None | NA | mRNA (Moderna), second | new | • AKI | 28 | High dose steroids | Responded to treatment at 1 month follow-up |
| | case 2 67 F | NA | NA | NA | mRNA (Moderna), second | relapse | • Oedema | 21 | High dose steroids, Rituximab | Responded to treatment at 2-month follow-up |
| Lim 2021 Korea | case 1 51 M | None | None | NA | Viral Vector (Janssen), first | new | • Oedema • Reduced urination • Weight gain | 7 | Furosemide, Methylprednisolone | Decreased serum creatinine, increased serum albumin after 7 days |
| Salem 2021 USA | case 1 33 F | None | MCD | NA | mRNA (Moderna), second | relapse | • Oedema • Headache • Vomiting • Hypertension | 21 | NA | NA |
| | case 2 41 F | Asthma | None | NA | mRNA (Pfizer), second | new | • Fever • Oedema • Weight gain • Hypertension | 5 | NA | NA |
| | case 3 34 F | None | MCD | NA | mRNA (Pfizer), second | relapse | • Oedema • Abdominal pain | 28 | NA | NA |
| Morlidge 2021 UK | case 1 30 M | None | MCD | NA | Viral Vector (Oxford-Astrazeneca), first | relapse | • Headache • Frothy urine | 2 | Prednisolone | Complete remission after 10 days treatment |
| | case 2 40 F | None | MCD | NA | Viral Vector (Oxford-Astrazeneca), first | relapse | • Headache • Frothy urine • Oedema | 1 | Prednisolone increased | Complete remission within 2 weeks |
| Özkan 2022 Turkey | case 1 33 F | None | MCD | NA | inactive SARS-CoV-2, second | relapse | • Foamy urine • Oedema | 14 | Methylprednisolone | NA |
| Kervella 2021 France | case 1 34 F | None | MCD | NA | mRNA (Pfizer), first | relapse | • Oedema | 10 | Increased corticosteroid dose | Complete remission after second dose relapse |
| Chandra 2022 USA | case 1 23 F | None | None | NA | mRNA (Moderna), second | new | • Oedema • Elevated blood pressure | 7 | Corticosteroids | Complete remission after 4 weeks |
| | case 2 74 M | Hypertension | None | NA | mRNA (Pfizer), second | new | • Oedema • Weight gain | 2 | Supportive therapy | Complete remission after ~6 weeks |
| | case 3 72 F | Hypertension, Obesity, Dyslipidaemia | None | None | Viral Vector (Oxford-AstraZeneca), First | new | • Oedema • Dyspnoea • Fatigue | 14 | Prednisolone | Complete remission of proteinuria and improved creatinine and albumin after 2 weeks treatment |
| | case 4 71 M | Acute myeloid leukaemia, Allogeneic hematopoietic stem cell transplantation, Glucocorticoid-induced diabetes, Mild GVHD in liver | MCD (GVHD) | NA | mRNA (Moderna), second | relapse | • Foamy and dark urine • Oedema • Abdominal bloating | 7 | Prednisolone, Rituximab, Loop diuretic | Complete remission after 7 months |

*(Continued)*

**Table 1.** (Continued)

| Author, Year, Country | Case number, Age, Gender | Comorbidities | Previous glomerulopathy | Previous COVID-19 infection | COVID-19 vaccine type and dose | New onset/ relapse of minimal change disease post vaccination | Main presenting symptoms | Days for the onset of symptoms | Treatment | Outcome |
|---|---|---|---|---|---|---|---|---|---|---|
| Hummel 2022 France | case 1 38 M | NA | NA | NA | Viral Vector (Oxford-AstraZeneca), first | relapse | NA | 14 | Corticosteroids, Mycophenolate Mofetil | Complete remission after 1 month |
| | case 3 74 M | NA | NA | NA | mRNA (Pfizer), first | relapse | NA | 21 | Corticosteroids, Calcineurin inhibitor | Complete remission after 3 months |
| | case 4 46 F | NA | NA | NA | mRNA (Pfizer), first | relapse | NA | 11 | Corticosteroids, Calcineurin inhibitor | Complete remission after 1 month |
| | case 5 23 M | NA | NA | NA | mRNA (Pfizer), first | relapse | NA | 21 | Corticosteroids, Obinutuzumab | Complete remission after 1 month |
| | case 6 30 F | NA | NA | NA | mRNA (Pfizer), second | relapse | NA | 6 | Corticosteroids, Rituximab | Complete remission after 1 month |
| | case 7 36 F | NA | NA | NA | mRNA (Pfizer), first | relapse | NA | 10 | Corticosteroids, Rituximab | Complete remission after 1 month |
| | case 8 41 F | NA | NA | NA | mRNA (Pfizer), first | relapse | NA | 30 | Corticosteroids, Calcineurin inhibitor | Complete remission after 1 month |
| | case 9 16 M | NA | NA | NA | mRNA (Pfizer), first | relapse | NA | 15 | Corticosteroids | Complete remission after 1 month |
| | case 10 19 M | NA | NA | NA | mRNA (Pfizer), first | relapse | NA | 21 | Corticosteroids | Complete remission after 1 month |
| | case 11 48 M | NA | NA | NA | mRNA (Moderna), first | relapse | NA | 7 | Corticosteroids, Mycophenolate mofetil | Complete remission after 1 month |
| | case 12 40 M | NA | NA | NA | mRNA (Pfizer), first | relapse | NA | 7 | Corticosteroids | Complete remission after 1 month |
| | case 14 83 M | NA | NA | NA | Viral Vector (Oxford-AstraZenecca), second | relapse | NA | 20 | Corticosteroids | Complete remission after 3 months |
| | case 15 53 F | NA | NA | NA | mRNA (Pfizer), first | relapse | NA | 26 | Corticosteroids | Complete remission after 1 month |
| | case 16 25 M | NA | NA | NA | mRNA (Pfizer), first | relapse | NA8 | 21 | Corticosteroids, Mycophenolate mofetil | Complete remission after 1 month |
| | case 17 19 M | NA | NA | NA | mRNA (Pfizer), second | relapse | NA | 25 | Corticosteroids | Complete remission after 1 month |
| | case 18 15 M | NA | NA | NA | mRNA (Pfizer), first | relapse | NA | 28 | Corticosteroids | Complete remission after 1 month |
| | case 19 31 M | NA | NA | NA | mRNA (Pfizer), first | relapse | NA | 21 | Corticosteroids | Complete remission after 1 month |
| | case 20 21 M | NA | NA | NA | mRNA (Pfizer), second | relapse | NA | 20 | Corticosteroids | Complete remission after 3 months |
| | case 21 42 M | NA | NA | NA | Viral Vector (Oxford-AstraZeneca), first | relapse | NA | 11 | Corticosteroids | Complete remission after 3 months |
| | case 22 72 M | NA | NA | NA | mRNA (Pfizer), third | relapse | NA | 7 | Corticosteroids, Mycophenolate mofetil | NA |
| | case 23 18 F | NA | NA | NA | mRNA (Pfizer), first | relapse | NA | 14 | Corticosteroids, Mycophenolate mofetil | Complete remission after 1 month |
| | case 24 16 F | NA | NA | NA | mRNA (Moderna), second | relapse | NA | 1 | Corticosteroids | Complete remission after 1 month |
| | case 25 72 M | NA | NA | NA | mRNA (Pfizer), third | relapse | NA | 2 | Corticosteroids | NA |
| Güngör 2022 Turkey | case 1 17 F | No | Idiopathic nephrotic syndrome | NA | modRNA, second | relapse | • Oedema | 19 | Corticosteroids | Remission after 2 weeks of treatment |
| | case 2 17.5 F | No | Idiopathic nephrotic syndrome | NA | NA, second | relapse | • Oedema | 12 | Corticosteroids | Remission after 2 weeks of treatment |

(*Continued*)

**Table 1.** (Continued)

| Author, Year, Country | Case number, Age, Gender | Comorbidities | Previous glomerulopathy | Previous COVID-19 infection | COVID-19 vaccine type and dose | New onset/ relapse of minimal change disease post vaccination | Main presenting symptoms | Days for the onset of symptoms | Treatment | Outcome |
|---|---|---|---|---|---|---|---|---|---|---|
| Fenoglio 2022 Italy | case 5 36 M | NA | No | NA | mRNA (Pfizer), second | new | • Urinary abnormalities | 82 | Rituximab | NA |
| | case 7 82 M | NA | No | NA | mRNA (Moderna), second | new | • Renal failure • Nephrotic syndrome | 79 | Glucocorticoids | NA |
| | case 8 54 F | NA | No | NA | mRNA (Moderna), second | new | • Nephrotic syndrome | 62 | Glucocorticoids | NA |
| | case 12 42 F | NA | No | NA | mRNA (Pfizer), second | new | • Renal failure • Nephrotic syndrome | 88 | MC | NA |
| | case 16 20 M | NA | No | NA | mRNA (Pfizer), first | new | • Nephrotic syndrome | 46 | Rituximab | NA |
| Lim 2022 Korea | case 2 52 M | No | No | NA | Viral Vector (Janssen), first | new | • Oedema • Nephrotic syndrome • Weight gain | 7 | Prednisolone | Complete response at 31 weeks |
| Dormann 2021 Germany | case 1 78 M | Arterial hypertension, Coronary heart disease, Hyperlipoproteinemia, COPD, Allergies | No | none | mRNA (Pfizer), first | new | • Oedema • Weight gain | 4 | Diuretics | Relapse after second dose *(see row below)* |
| | case 1 (2) 78 M | *(See row above)* | *(See row above)* | *(See row above)* | second | relapse | • Oedema • Weight gain • Pleural effusion | 14 | Prednisolone, Diuretics, Anticoagulants | Partial remission, reduced proteinuria and weight loss after 3 weeks |
| | case 2 31 F | Lipedema | No | none | Viral Vector (Janssen), first | new | • Oedema • Foamy urine • Syncope with orthostatic dysregulation | 0 | Prednisolone, Antibiotics, Immunoglobulin, Rituximab, Anticoagulation, Diuretics | Complete remission with mild hyperlipoproteinemia at day 52 |
| Anupama 2021 India | case 1 19 F | NA | No | NA | Viral Vector (Oxford-AstraZeneca), first | new | • Oedema | 8 | Prednisolone | Clinical and biochemical remission |
| Schwotzer 2021 Switzerland | case 1 22 M | No | MCD | NA | mRNA (Pfizer), NA | relapse | • Chills and low-grade fever • Proteinuria | 2 | Prednisolone, Tacrolimus | Remission after 17 days treatment |
| Hong 2022 Taiwan | case 1 51 M | No | No | NA | mRNA (Moderna), second | new | • Oedema • Foamy urine | 3 | Prednisolone, Angiotensin 2 receptor blocker | Complete remission at 10 weeks treatment |
| Timmermans 2022 Netherlands | case 1 64 F | NA | No | none | Viral Vector (Oxford-AstraZeneca), first | new | • Oedema | 7 | Prednisolone | Complete remission after 4 months |
| | case 2 34 M | NA | No | none | mRNA (Pfizer), second | new | NA | 28 | No | NA |
| | case 3 74 M | NA | No | none | mRNA (Pfizer), second | new | NA | 42 | Prednisolone | NA |
| Nakazawa 2022 Japan | case 1 15 M | No | No | yes | mRNA (Pfizer), first | new | • Oedema • Weight gain | 4 | Prednisolone | Complete remission at 12 days of treatment |
| Arias 2022 Spain | case 1 28 F | No | Idiopathic nephrotic syndrome | yes | Viral Vector (Oxford-AstraZeneca), first | relapse | • Oedema | 2 | Prednisolone, Atorvastatin, Antiplatelet therapy, Omeprazole, Trimethoprim-sulfamethoxazole | Negative proteinuria and no oedema after 4 weeks of treatment |
| Haider 2022 Italy | case 1 63 M | No | MCD | NA | mRNA (Pfizer), booster | relapse | • Oedema • Weight gain • Elevated blood pressure | < 7 | Prednisolone | Normal protein:creatinie ratio after 2 weeks treatment |
| Fehr 2021 Switzerland | case 1 65 M | Collagenous colitis | No | NA | mRNA (Moderna), first | new | • Nephrotic syndrome • AKI | 8 | Dialysis, Immunosuppressive therapy | Complete remission after treatment |

*(Continued)*

**Table 1.** (Continued)

| Author, Year, Country | Case number, Age, Gender | Comorbidities | Previous glomerulopathy | Previous COVID-19 infection | COVID-19 vaccine type and dose | New onset/ relapse of minimal change disease post vaccination | Main presenting symptoms | Days for the onset of symptoms | Treatment | Outcome |
|---|---|---|---|---|---|---|---|---|---|---|
| Nagai 2022 Japan | case 1 22 M | No | No | NA | mRNA (Pfizer), first | new | • Oedema | 9 | Heparin, Prednisolone, Furosemide | Clinical signs disappeared on 7th day of treatment |
| Caza 2021 USA | case 3 70 F | NA | No | none | mRNA (Pfizer), second | new | • AKI • Nephrotic syndrome | < 7 | Steroid therapy | Recovery at 4 weeks |
| | case 4 43 F | NA | No | none | mRNA (Pfizer), second | new | • Nephrotic syndrome | 14 | Steroid therapy | Recovery at 4 weeks |
| | case 5 79 M | NA | No | none | mRNA (NA), first | new | • AKI • Nephrotic syndrome | < 14 | Steroid therapy | Recovery at 4 weeks |
| | case 6 72 M | NA | No | none | mRNA (Moderna), second | new | Nephrotic syndrome | 7 | Steroid therapy, ACEi | Recovery at 2 weeks |
| | case 7 47 F | NA | No | none | mRNA (Pfizer), second | new | • AKI • Nephrotic syndrome | < 14 | Dialysis, Steroid therapy, ACEi | No recovery at 4 weeks |
| | case 8 23 M | NA | No | none | Viral Vector (Oxford-AstraZeneca), first | new | • AKI • Nephrotic syndrome | 14 | Steroid therapy, Dialysis | Recovery at 3 weeks |
| | case 9 45 F | NA | No | none | mRNA (Moderna), first | new | • Nephrotic syndrome | < 14 | Steroid therapy | NA |
| Fornara 2022 Italy | case 4 66 F | Hypertension TIA | No | NA | mRNA (Pfizer), second | new | NA | 160 | Oral steroids | Partial remission after 56 days |
| Leclerc 2021 Canada | case 1 71 M | Dyslipidaemia | No | NA | Viral Vector (Oxford-AstraZeneca), first | new | • Oedema • Elevated blood pressure • AKI | 1 | Methylprednisolone, Prednisolone, Haemodialysis | Improvement after 30 days treatment |
| Mancianti 2021 Italy | case 1 39 M | No | MCD | none in the weeks prior | mRNA (Pfizer), first | relapse | • Oedema • Fatigue • AKI | 3 | Prednisolone | Complete remission after 4 weeks treatment |
| Holzworth 2021 USA | case 1 63 F | Hypertension, Tobacco dependence | No | NA | mRNA (Moderna), first | new | • Oedema • Dyspnoea • Fatigue • Frothy urine • Elevated blood pressure • Mild AKI | less than 7 days | Methylprednisolone, Prednisolone, Valsartan, Loop diuretic | NA |
| Komaba 2021 Japan | case 1 60s M | No | MCD | NA | mRNA (Pfizer), first | relapse | • Frothy urine | 8 | Prednisolone, Cyclosporine | Proteinuria resolved within 2 weeks treatment |
| D'Agati 2021 USA | case 1 77 M | Type 2 diabetes mellitus, Coronary artery disease, Obesity | No | NA | mRNA (Pfizer), first | new | • Oedema • Weight gain • Elevated blood pressure • Proteinuria | 7 | Methylprednisolone, Prednisolone, Furosemide, Bumetanide | No improvement after 3 weeks treatment |
| Maas 2021 Netherlands | case 1 80s M | Venous thromboembolism | No | NA | mRNA (Pfizer), first | new | • Oedema • Weight gain | 7 | Prednisolsteone | Improvement after 10 days treatment |

ACEi: angiotensin converting enzyme inhibitor, AKI: acute kidney injury, F: female, GVHD: graft versus host disease, M: male, MCD: Minimal Change Disease, NA: not available

achieved recovery or improved following treatment (19.1%, 18/94). Two people relapsed after treatment (2.1%, 2/94) and two cases (2.1%, 2/94) were reported as not recovered. In 15 cases (16%, 15/94) data about outcome was not provided.

## Quality of the studies

The mean quality score indicated that the studies reported on average 10 of the recommended 12 elements, defined by the guidelines. Ten studies had a perfect score of 12 while the second most common score was 11 (S2 Table).

## Discussion

The administration of COVID-19 vaccines has not been deemed as a causative factor for kidney disease. However, recent findings, primarily derived from case reports and case series, indicate that various kidney disorders such as Minimal Change Disease (MCD), IgA nephropathy, membranous glomerulopathy, and IgG4-related disease have been observed to initially manifest or relapse subsequent to SARS-CoV-2 vaccination. These observations suggest a potential link between COVID-19 vaccination and the occurrence or recurrence of MCD. In this study, we conducted a thorough screening of the existing literature to present a comprehensive summary of documented cases of MCD following SARS-CoV-2 vaccination. Our systematic review identified 46 relevant reports, involving a total of 93 patients, in which MCD was observed subsequent to the administration of various COVID-19 vaccines. In the majority of cases, symptoms began to emerge following the first vaccine dose, and clinical improvement was reported for most patients.

### Results in the context of the literature

MCD consists the most frequent cause of nephrotic syndrome in childhood and rarely affects adults. MCD generally presents with a sudden onset of symptoms and signs of nephrotic syndrome and requires histologic confirmation in adults. Its pathogenesis remains to be elucidated, however, evidence points towards T cell dysfunction being a major mechanism [66]. It has been previously proposed that a glomerular permeability factor is produced, attacks the glomerular filtration barrier and leads to the destruction of podocytes and subsequent proteinuria. It's most commonly idiopathic, but infections, medications, vaccinations, malignancies, and allergens are among the secondary etiologic factors [67]. Infections including syphilis, hepatitis C and tuberculosis, and vaccinations against hepatitis B, influenza, measles and rubella are established triggering factors for the relapse of primary glomerulonephritis, potentially with a similar mechanism involved in the development of MCD [68].

In animal models the prevalence of CD8+ suppressor T-cells and subsequent cytokine-induced injury has been observed in the active phase of MCD [60, 69]. This could provide a possible explanation for the aforementioned cases since the existent vaccinations against COVID-19 are known to strongly induce T-cell activation and this could lead to immune mediated podocyte damage. It's worth noting that during the vaccine-induced T-cell activation, interferon gamma and inerleukin-2 (IL-2) are increased and IL-2 has been found to be raised in the acute phase and relapses of idiopathic nephrotic syndrome [22]. Direct podocyte injury could also be implicated in MCD in both COVID-19 infection and vaccination and interestingly ACE-2 is expressed in podocytes, however there is currently not adequate evidence to establish a causative mechanism. Moreover, similarities between vaccine adjuvants and human proteins could lead to immune cross-reactivity and drug-induced hypersensitivity reactions through molecular mimicry [70, 71].

Even though MCD most commonly presents during childhood, it has been reported mainly in adults following COVID-19 vaccination, however this is to be expected considering the high vaccination rates in these age groups. MCD symptomatology commenced within 3 weeks from the first dose in more than half of the patients, although a significant amount of people developed symptoms after the second dose, which could be associated to the amplitude of the immune response. Symptoms did not differ from those commonly reported in literature and glucocorticoids were chosen as first-line treatment in 91.5% of the cases. Concerns about potential interference of immunosuppressive agents such as rituximab in the vaccination efficacy has been raised, however, relevant treatment to achieve best clinical response should be prioritised over immunisation in these cases. Overall the vast majority responded to treatment and maintained positive outcomes.

### Strengths and limitations

Our study represents the first systematic review conducted on the relationship between COVID-19 vaccination and the occurrence or relapse of MCD. Our findings present a comprehensive overview of published reports with quality assessment of the included studies.

However, it is important to highlight certain limitations linked to our study. One major limitation stems from the low quality nature of the case reports and case series included in this review, which can impact the validity and generalizability of the conclusions. These studies are susceptible to potential biases including overinterpretation and selection bias. Consequently, while the reported findings are interesting, they may not necessarily provide an accurate representation of the true effect of COVID-19 vaccination in relation to renal dysfunction. Therefore, establishing causality requires insight from mechanistic studies and well-designed appropriately powered prospective studies.

### Conclusion

While the current COVID-19 vaccines are generally considered safe and the advantages of vaccination outweigh the potential occurrence of adverse events, it is possible for patients to develop mild to moderate side effects, including complications related to renal dysfunction. Minimal change disease is possibly a condition clinical doctors and other healthcare professionals may expect to see in patients receiving COVID-19 vaccines. Although this adverse event is uncommon, considering the limited published data and the absence of confirmed causality, increased clinical awareness is crucial for the early recognition and optimal management of these patients.

### Supporting information

**S1 Table. Laboratory results and imaging findings for the included cases.**
(DOCX)

**S2 Table. Quality assessment of the included studies.**
(DOCX)

**S1 Fig. Prisma checklist.**
(DOC)

### Author Contributions

**Conceptualization:** Konstantinos S. Kechagias, Dimitra Stathi.

**Data curation:** Konstantinos S. Kechagias, Joshua D. Laleye, Jan Drmota, Georgios Kyrtsonis, Marina Zafeiri, Konstantinos Katsikas Triantafyllidis, Dimitra Stathi.

**Formal analysis:** Konstantinos S. Kechagias, Joshua D. Laleye, Jan Drmota, Georgios Kyrtsonis, Marina Zafeiri, Konstantinos Katsikas Triantafyllidis, Dimitra Stathi.

**Investigation:** Konstantinos S. Kechagias, Joshua D. Laleye, Jan Drmota, Georgios Kyrtsonis, Marina Zafeiri, Konstantinos Katsikas Triantafyllidis, Dimitra Stathi.

**Methodology:** Konstantinos S. Kechagias, Joshua D. Laleye, Jan Drmota, Georgios Geropoulos, Georgios Kyrtsonis, Marina Zafeiri, Konstantinos Katsikas Triantafyllidis, Dimitra Stathi.

**Project administration:** Konstantinos S. Kechagias, Joshua D. Laleye, Marina Zafeiri, Dimitra Stathi.

**Resources:** Konstantinos S. Kechagias, Georgios Geropoulos, Georgios Kyrtsonis, Marina Zafeiri, Konstantinos Katsikas Triantafyllidis, Dimitra Stathi.

**Software:** Konstantinos S. Kechagias, Dimitra Stathi.

**Supervision:** Konstantinos S. Kechagias, Marina Zafeiri, Konstantinos Katsikas Triantafyllidis, Dimitra Stathi.

**Validation:** Konstantinos S. Kechagias, Joshua D. Laleye, Jan Drmota, Georgios Geropoulos, Georgios Kyrtsonis, Marina Zafeiri, Konstantinos Katsikas Triantafyllidis, Dimitra Stathi.

**Visualization:** Konstantinos S. Kechagias, Georgios Geropoulos, Marina Zafeiri, Konstantinos Katsikas Triantafyllidis, Dimitra Stathi.

**Writing – original draft:** Konstantinos S. Kechagias, Joshua D. Laleye, Jan Drmota, Georgios Geropoulos, Georgios Kyrtsonis, Marina Zafeiri, Konstantinos Katsikas Triantafyllidis, Dimitra Stathi.

**Writing – review & editing:** Konstantinos S. Kechagias, Georgios Geropoulos, Dimitra Stathi.

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
