## [Decision Letter · Decision Letter 0]

3 Sep 2023

PONE-D-23-13443Minimal change disease following COVID-19 vaccination a systematic reviewPLOS ONE

Dear Dr.Konstantinos S. Kechagias,

Thank you for submitting your manuscript to PLOS ONE. After careful consideration, we feel that it has merit but does not fully meet PLOS ONE’s publication criteria as it currently stands. Therefore, we invite you to submit a revised version of the manuscript that addresses the points raised during the review process.

We look forward to receiving your revised manuscript.

Kind regards,

Rajendra Bhimma, PhD

Academic Editor

PLOS ONE

Journal Requirements:

- https://rcastoragev2.blob.core.windows.net/09e270450a896d88e5d0c14e0f588a12/PMC9552880.pdf

In your revision ensure you cite all your sources (including your own works), and quote or rephrase any duplicated text outside the methods section. Further consideration is dependent on these concerns being addressed.

3. Please include a caption for figure 1. 

Additional Editor Comments:

Thank you for your submisson of the manuscript "Minimal change disease following COVID-19 vaccination a systematic review". Three reviewers have raised concerns and these need to be addressed.

Reviewers' comments:

Reviewer's Responses to Questions

**Comments to the Author**

1. Is the manuscript technically sound, and do the data support the conclusions?

Reviewer #1: No

Reviewer #2: Yes

Reviewer #3: Yes

2. Has the statistical analysis been performed appropriately and rigorously? 

Reviewer #1: No

Reviewer #2: I Don't Know

Reviewer #3: I Don't Know

3. Have the authors made all data underlying the findings in their manuscript fully available?

Reviewer #1: Yes

Reviewer #2: Yes

Reviewer #3: Yes

4. Is the manuscript presented in an intelligible fashion and written in standard English?

Reviewer #1: Yes

Reviewer #2: Yes

Reviewer #3: Yes

5. Review Comments to the Author

Reviewer #1: Please see the updated articles on various adverse effect of astrazeneca and other mrna covid vaccine.Authors missed the major complication such as sinus venous thrombosis as well as inflammatory cns disorders post vaccination and other relevant findings.

Also the information can be obrained from Medicines and Healthcare products Regulatory Agency (MHRA) data as well from Vaccine adverse events reporting system

Reviewer #2: The research question is clear. The inclusion and exclusion criteria are well defined. The search strategy, study selection and data extraction appear comprehensive. Data synthesis is appropriate.

The interpretation of the findings and conclusion are accurate.

The following revisions are required:

Line 29 change “renal dysfunction” to more appropriate term – *Glomerular disease or chronic kidney disease

Line 45 change terms – “into remission “

Line 49 change “clinical doctors” to clinicians in the paper

Line 91 correct spelling of vaccine

Line 147/8 biopsy reported in only 71% of cases ( ? diagnosis) – important limitation , if these were relapses then a second biopsy would be unlikely , although first time diagnosis would require a biopsy , specify if primary or relapse.

First line of discussion “renal dysfunction” term should be changed

Second line of paragraph : Results in context of literature:

“Evidence points towards T cell dysfunction being a major Mechanism” Is this immune dysregulation with dysfunction of T regulatory cells as with other cases of MCD?

Line 3-4 “glomerular capillary wall” could be restated as the glomerular filtration barrier.

Last line “requires histologic confirmation in adults” is fundamental to diagnosis. MCD is a histological diagnosis. Unless there was relapse which did not warrant a biopsy, then all other cases would require confirmatory histology. If this is not so, then it should be listed as a limitation of the review.

Reviewer #3: It is a relevant topic with billions of doses of vaccines used in such a short period of time and also Being RNA based.

1. Need clarity on that only 76% had histological classification of their nephrosis. what criteria were used to establish minimal change.

2. Was there any particular vaccine type that caused the most or least number of nephrotic syndrome cases. I know that then we have to look at the number of vaccine doses but was there a particular type involved with the new cases compared to the relapses?

3. There are 2 diabetics with a possibility of diabetic neohropathy? Was minimal change confirmed on histology?

4. The other secondary involvement were cardiac problems such as hypertension, hyperlipidaemia, and arrthytmias

althoiugh hypertension and hyperlipdaemis can be caused by the nephrosis

5. If only 76% had renal histology, which criteria were used to prove minimal change?

6. Infections that were mentioned in the discussion such as Hep 'B' cause membranous nephropathy.

It is well written. Just the above queries or clarification.

6. PLOS authors have the option to publish the peer review history of their article (what does this mean?). If published, this will include your full peer review and any attached files.

Reviewer #1: **Yes: **shitiz sriwastava

Reviewer #2: No

Reviewer #3: No

---

## [Author Response · Author response to Decision Letter 0]

30 Oct 2023

Dear Editor,

Thank you for considering our article ‘Minimal change disease following COVID-19 vaccination: a systematic review’ for publication in your journal. 

We enclose below a point-by-point response to the reviewers’ comments and a revised version of the manuscript. We have also adjusted the format according to the journal’s guidance.

We look forward to your reply.

Yours sincerely

Konstantinos Kechagias MD, MSc

Clinical research fellow 

Institute of Reproductive and Developmental Biology, Department of Metabolism Digestion and Reproduction, Imperial College London

Reviewer #1: Please see the updated articles on various adverse effect of astrazeneca and other mrna covid vaccine. Authors missed the major complication such as sinus venous thrombosis as well as inflammatory cns disorders post vaccination and other relevant findings.

Also the information can be obtained from Medicines and Healthcare products Regulatory Agency (MHRA) data as well from Vaccine adverse events reporting system.

Thank you for the comment. Although different types of adverse events secondary to Covid-19 vaccine have been recognized, the focus of the current literature review was minimal change disease. In the introduction section of the article, we briefly mention the plethora of other adverse events related to Covid-19 vaccines and we have now also included the MHRA guidance as reference. 

Line 72. 

 

Reviewer #2: The research question is clear. The inclusion and exclusion criteria are well defined. The search strategy, study selection and data extraction appear comprehensive. Data synthesis is appropriate.

The interpretation of the findings and conclusion are accurate.

The following revisions are required:

Line 29 change “renal dysfunction” to more appropriate term – *Glomerular disease or chronic kidney disease. Line 45 change terms – “into remission “

Line 49 change “clinical doctors” to clinicians in the paper

Line 91 correct spelling of vaccine. 

We thank the reviewer for the annotations. The above have now been changed as per recommendations. Regarding line 91, the term vaccin* was used in order to identify articles including words such as vaccine, vaccines and vaccination. 

Lines 29, 45 and 49.

Line 147/8 biopsy reported in only 71% of cases (? diagnosis) – important limitation, if these were relapses then a second biopsy would be unlikely, although first time diagnosis would require a biopsy, specify if primary or relapse.

We thank the reviewer for the comment. We elaborated on the use of biopsy as a diagnostic tool. Biopsy reports were provided for 67 cases (71.2%, 67/94) and were consistent with MCD. MCD was confirmed with biopsy in 76 cases (80.8%, 76/94). Sixteen cases (17%, 16/94) were a relapse and biopsy was not repeated. In two cases (2.1%), diagnosis was based on clinical suspicion (Nagai et al, Nakazawa et al).

Lines 147 and 148. 

First line of discussion “renal dysfunction” term should be changed. 

Thank you for the annotation. Renal dysfunction has been replaced by kidney disease. 

Line 161

Second line of paragraph: Results in context of literature:

“Evidence points towards T cell dysfunction being a major Mechanism” Is this immune dysregulation with dysfunction of T regulatory cells as with other cases of MCD?

We thank the reviewer for this comment. Yes, this refers to the dysfunction of T regulatory cells and the hypothesis that a glomerular permeability factor is produced that subsequently attacks the glomerular membrane, which has been proposed in the past as a potential mechanism for MCD. This has now been clarified in the manuscript. 

Line 175 

Line 3-4 “glomerular capillary wall” could be restated as the glomerular filtration barrier.

Thank you for the annotation. This has now been restated as suggested. 

Lines 3 and 4. 

Last line “requires histologic confirmation in adults” is fundamental to diagnosis. MCD is a histological diagnosis. Unless there was relapse which did not warrant a biopsy, then all other cases would require confirmatory histology. If this is not so, then it should be listed as a limitation of the review.

We thank the reviewer for the comment, and we agree. Biopsy reports were provided for 67 cases (71.2%, 67/94) and were consistent with MCD. MCD was confirmed with biopsy in 76 cases (80.8%, 76/94). Sixteen cases (17%, 16/94) were relapses and biopsy was not repeated. In two cases (2.1%), diagnosis was based on clinical suspicion.

Lines 147-149

 

Reviewer #3: It is a relevant topic with billions of doses of vaccines used in such a short period of time and also Being RNA based.

1. Need clarity on that only 76% had histological classification of their nephrosis. what criteria were used to establish minimal change.

We thank the reviewer for the comment and we agree. Biopsy reports were provided for 67 cases (71.2%, 67/94) and were consistent with MCD. MCD was confirmed with biopsy in 76 cases (80.8%, 76/94). Sixteen cases (17%, 16/94) were a relapse and biopsy was not repeated. In two cases (2.1%), diagnosis was based on clinical suspicion (Nagai, Nakazawa).

Lines 147-149

2. Was there any particular vaccine type that caused the most or least number of nephrotic syndrome cases. I know that then we have to look at the number of vaccine doses but was there a particular type involved with the new cases compared to the relapses?

We thank the reviewer for the comment. In the results section, both the vaccine type and the number of doses before the development of nephrotic syndrome are mentioned. The majority of the patients received COMIRNATY (58.5%, 55/94), followed by COVID-19 Vaccine Moderna (20.2%, 19/94) and VAXZEVRIA (14%, 13/94), while 4 participants received COVID-19 Vaccine Janssen (3.2%, 3/94) and CoronaVac (1%, 1/94). In one case vaccine type was reported as modRNA (1%, 1/94). The majority of patients developed symptoms after the first dose (55.5%, 52/94), followed by the second dose (39.3%, 37/94), third dose (2.2%, 2/94), booster (1%, 1/94), both first and second doses (1%, 1/94), while in one case relevant data was not provided (1%, 1/94). 

Lines 138-143

3. There are 2 diabetics with a possibility of diabetic nephropathy? Was minimal change confirmed on histology? The other secondary involvement were cardiac problems such as hypertension, hyperlipidaemia, and arrthytmias although hypertension and hyperlipdaemis can be caused by the nephrosis

Thank you for your comment. Patients with the above comorbidities had histological diagnosis of MCD. 

5. If only 76% had renal histology, which criteria were used to prove minimal change?

We thank the reviewer for this comment. Biopsy reports were provided for 67 cases (71.2%, 67/94) and were consistent with MCD. MCD was confirmed with biopsy in 76 cases (80.8%, 76/94). Sixteen cases (17%, 16/94) were relapses and biopsy was not repeated. In two cases (2.1%), diagnosis was based on clinical suspicion.

Lines 147-149

6. Infections that were mentioned in the discussion such as Hep 'B' cause membranous nephropathy.

We thank the reviewer for this comment. We mentioned Hep B and other viruses as triggering factors for primary glomerulonephrosis including MCD given that a similar pathophysiological mechanism could be implicated.

---

## [Decision Letter · Decision Letter 1]

9 Jan 2024

Minimal change disease following COVID-19 vaccination a systematic review

PONE-D-23-13443R1

Dear Dr. Kostantionos S Kechagias

We’re pleased to inform you that your manuscript has been judged scientifically suitable for publication and will be formally accepted for publication once it meets all outstanding technical requirements.

Kind regards,

Rajendra Bhimma, PhD

Academic Editor

PLOS ONE

Additional Editor Comments (optional):

Reviewers' comments:

Reviewer's Responses to Questions

**Comments to the Author**

1. If the authors have adequately addressed your comments raised in a previous round of review and you feel that this manuscript is now acceptable for publication, you may indicate that here to bypass the “Comments to the Author” section, enter your conflict of interest statement in the “Confidential to Editor” section, and submit your "Accept" recommendation.

Reviewer #2: All comments have been addressed

Reviewer #3: All comments have been addressed

2. Is the manuscript technically sound, and do the data support the conclusions?

Reviewer #2: Yes

Reviewer #3: Yes

3. Has the statistical analysis been performed appropriately and rigorously? 

Reviewer #2: I Don't Know

Reviewer #3: I Don't Know

4. Have the authors made all data underlying the findings in their manuscript fully available?

Reviewer #2: Yes

Reviewer #3: Yes

5. Is the manuscript presented in an intelligible fashion and written in standard English?

Reviewer #2: Yes

Reviewer #3: Yes

6. Review Comments to the Author

Reviewer #2: The authors have indicated that they will amend the paper according to each point raised. If this is done completely then they have complied with the review.

Reviewer #3: It is a relevant review ion that the complication of nephrotic syndrome or relapse is an infrequent occurrernce of Covid vaccination.The practising physicians need to know of the renal complications of Covid vaccination. Also alert the researchers to do a prospective studies as regards incidence, severity, associated factors, type of vaccine etc. This was a systemic review of the literature, where their is no uniformity of the reported information and almost a third were case reports.

In my opinion it is a relevant review to spark prospective studies.

7. PLOS authors have the option to publish the peer review history of their article (what does this mean?). If published, this will include your full peer review and any attached files.

Reviewer #2: No

Reviewer #3: No

---

## [Editor Report · Acceptance letter]

24 Feb 2024

PONE-D-23-13443R1 

PLOS ONE

Dear Dr. Kechagias, 

I'm pleased to inform you that your manuscript has been deemed suitable for publication in PLOS ONE. Congratulations! Your manuscript is now being handed over to our production team.

Kind regards, 

on behalf of

Professor Rajendra Bhimma 

Academic Editor

PLOS ONE